# Decision-Based Fusion for Vehicle Matching

**DOI:** 10.3390/s22072803

**Published:** 2022-04-06

**Authors:** Sally Ghanem, Ryan A. Kerekes, Ryan Tokola

**Affiliations:** 1Oak Ridge National Laboratory, Oak Ridge, TN 37830, USA; kerekesra@ornl.gov; 2Aware, Inc., Bedford, MA 01730, USA; ryantokola@gmail.com

**Keywords:** decision fusion, deep networks, vehicle matching

## Abstract

In this work, a framework is proposed for decision fusion utilizing features extracted from vehicle images and their detected wheels. Siamese networks are exploited to extract key signatures from pairs of vehicle images. Our approach then examines the extent of reliance between signatures generated from vehicle images to robustly integrate different similarity scores and provide a more informed decision for vehicle matching. To that end, a dataset was collected that contains hundreds of thousands of side-view vehicle images under different illumination conditions and elevation angles. Experiments show that our approach could achieve better matching accuracy by taking into account the decisions made by a whole-vehicle or wheels-only matching network.

## 1. Introduction

Data fusion is a very challenging topic in machine learning that can provide complimentary information and enhance the understanding of data structures. Multi-modal data have become more widely available due to advances in sensor development. While data fusion is not a new topic, it has recently witnessed a considerable increase in the demand for foundational principles for different applications. There are three levels of data fusion that can be considered, namely, data fusion, feature fusion, and decision fusion. Data fusion involves integrating unprocessed data generated by each sensor. In feature fusion, features extracted from raw data, collected from various sensors, and combined to construct a better feature. Decision fusion, on the other hand, optimally combines the decisions reached by different algorithms or sensors to yield a more informed decision.

The objective of this work was to develop a principled decision fusion framework for vehicle matching using convolutional neural networks (CNNs). The topic of vehicle re-identification has been studied extensively in the literature. In [1], the authors proposed a local feature-aware model for vehicle re-identification. Given multi-view images of a target vehicle, their model focuses on learning informative parts that are most likely to differ among vehicles. However, their model does not perform well on images with a dim backgrounds. Additionally, the model can not achieve effective identification in the case of only different views of two cars or in the absence of shared parts in the two cars. In [2,3], the authors adopted a spatio-temporal approach for vehicle re-identification. Bing et al. [4] proposed a part-regularized discriminative feature preserving method that enhances the ability to perceive subtle discrepancies. Their model utilized three vehicle parts for detection: lights, including front light and back light; window, including front window and back window; and vehicle brand. Oliveira et al. [5] presented a two-stream Siamese neural network that used both the vehicle shape and license plate information for vehicle re-identification. In [6], the authors proposed an end-to-end RNN-based hierarchical attention (RNN-HA) classification model for vehicle re-identification. Their RNN-based module models the coarse-to-fine category hierarchical dependency to effectively capture subtle visual appearance cues, such as customized paint and windshield stickers.

In this work, key features from vehicle side-view images and their corresponding wheels are extracted using Siamese networks. Pattern information specific to the wheels can often provide supplementary information about the vehicles in question; for example, two otherwise identical vehicles of the same make, model, and color, could potentially be distinguished if their wheels or hubcaps are different. Our dataset was collected under various illumination conditions and elevation angles. Individual similarity scores are combined, which are derived from features extracted from either the whole-vehicle images or from cropped images of their wheels. A principled integration of the individual scores is expected to enhance overall matching accuracy; thus, an overall similarity score is reached by a joint aggregation of the whole-vehicle and wheel similarity scores.

The balance of the paper is organized as follows. In Section 2, related work is described. In Section 3, the dataset structure is thoroughly explained. In Section 4, the attributes of our approach are provided and the network structure is defined. In Section 5, our validation along with other experimental results are presented, and Section 6 provides concluding remarks and describes future work.

## 2. Related Work

The topic of multi-modal data fusion has been extensively studied in computer vision. Laying out the fundamentals for data fusion has become crucial for many applications, including target recognition [7,8,9], handwriting analysis [10], and image fusion [11]. A comprehensive survey of data fusion is provided in [12,13].

Decision fusion techniques can be classified on the basis of the fusion type. The most popular fusion type is voting-based, which includes majority voting, weighted voting, and Borda count, which sums the reverse ranks to perform decision fusion [14]. Other voting techniques are probability-based, such as Bayesian Inference [15] and Dempster–Shafer fusion [16,17]. A detailed comparison of Bayesian inference and Dempster–Shafer fusion is included in [18]. A limitation common to probability based methods is they typically require prior information about sensors’ decisions or demand high computational complexity. As a result, their adoption in decision fusion in real-time applications has been negatively impacted. In this work, a decision fusion approach is established that leverages a neural network structure to mitigate the need for prior knowledge or assumptions about the classifiers. The performance of our proposed approach is then compared to that of the better-known majority vote fusion method.

Developing real-time transportation technologies remains a crucial topic for safety, surveillance, security, and robotics. Decisions regarding traffic flow and surveillance need to be performed in real time to detect potential threats and act accordingly. Computer vision systems can be employed to automatically match and track vehicles of interest. In [19], the authors developed a robust framework for matching vehicles with highly varying poses and illumination conditions. Moreover, in [20], the authors proposed a low complexity method for vehicle matching that is robust to appearance changes and inaccuracies in vehicle detection. They represented vehicle appearances using signature vectors and compared them using a combination of 1D correlations. A vehicle matching algorithm was proposed in [21] that identified the same vehicle in different camera sites using color information. Their matching approach took advantage of road color variation to model changes in illumination to compensate for color variation and minimize the false positive matches.

The primary contribution of this paper is a decision fusion framework for vehicle matching which aggregates decisions from two Siamese networks handling a pair of vehicle images and their detected wheels. Integrating the decisions helps reinforce the consistency between the outputs of the matching networks. In our evaluation, a recently collected dataset was used called Profile Images and Annotations for Vehicle Re-identification Algorithms (PRIMAVERA) [22], which has been made publicly available. These data were partitioned into training and validation subsets. After training the vehicle and the wheel matching networks, the learned networks were then utilized to match new observed data and investigate the generalization power of our approach. Experimental results confirmed a significant improvement in the vehicle matching accuracy under decision fusion.

## 3. Dataset Description

### 3.1. Data Collection

To substantiate the validation of our proposed approach, a dataset that contains hundreds of thousands of side-view vehicle images was collected. As mentioned above, this dataset has been made publicly available [22]. The data were collected using a roadside sensor system containing one or more color cameras and a radar unit. Three types of cameras were used for vehicle image collections. For daytime image capture, RGB cameras equipped with either Sony IMX290 (1945 × 1097 pixels) or IMX036 (2080 × 1552 pixels) sensors were used. For low-light image capture after sunset, a Sony UMC-S3C camera was used to perform high-sensitivity RGB imaging. Images were captured from distances to the vehicle ranging between 1 and 20 m using 1.8 to 6 mm lenses in conjunction with the above cameras. An undistortion operation was applied to each frame prior to any processing in order to remove distortion effects of the wide-angle lenses. Images of passing vehicles were collected over the course of several years. The images that we used in this study were captured during both day and night. The nighttime imagery was captured using a low-light color camera. The sensors were positioned both at ground level and at an elevated angle of approximately 20 degrees from horizontal and were oriented perpendicular to the road, providing a near-profile view of passing vehicles.

License plate readers were collocated with the sensors to provide a ground-truth identity for each collected vehicle. While license plates were used for ground truth, this approach to vehicle re-identification does not rely on a license plate but only a profile view of the vehicle. Actual license plate numbers were obfuscated by replacing each plate number with an arbitrary number that was subsequently used as the vehicle ID. Sample images from the dataset are depicted in Figure 1.

### 3.2. Data Quantity

The dataset contains 636,246 images representing 13,963 vehicles. Each vehicle has a different number of images depending on the number of collected video frames and how many times it passed by one of the sensors. Our dataset was divided into training and validation sets. The training set contains 543,926 images representing 11,918 vehicles, and the validation set contains 92,320 images representing 2045 vehicles.

## 4. Network Structure

Our vehicle matching algorithm consists of two main stages. The first stage includes a whole-vehicle matching network, which compares a pair of vehicle images and generates a similarity score. The first stage also contains a wheel detector, which detects the outermost two wheels in each vehicle image and feeds them into the wheel matching network. This network compares the detected wheels and outputs wheel similarity scores. The second stage of the proposed framework is the decision fusion network, which combines the vehicle and wheel matching scores toward the goal of obtaining a more accurate and robust overall similarity score. The diagram of the proposed framework is depicted in Figure 2.

### 4.1. Problem Formulation

In the following, our problem formulation is explained. Assume having two data points, $X_{i}^{t}$ and $X_{\mathrm{ii}}^{t}$
$\in R^{m\times n}$ that belong to class *i* and ii, respectively. In addition, consider having *T* different modalities such that *t* denotes the modality index and t=1,…,T. More precisely, consider two feature maps $x_{i}^{t}$ and $x_{\mathrm{ii}}^{t}$
$\in R^{c}$ corresponding to $X_{i}^{t}$ and $X_{\mathrm{ii}}^{t}$, respectively. For each modality *t*, a decision $D^{t}$ is generated such that $D^{t}=f\left(x_{i}^{t},x_{\mathrm{ii}}^{t}\right)$, where *f* is the matching function and $D^{t}$
∈R. Our goal is to fuse the decisions $D_{t=1}^{T}$ so a fusion function k:D(1),D(2)…,D(T)→z is defined that fuses the decisions $D_{t=1}^{T}$ and produces a final decision *z*
∈R. Various fusion functions can be used to combine the matching decisions. In this work, the use of neural networks was investigated to better explore the potential contribution of decision fusion. A multi-layered, fully connected decision fusion network was trained that combines the matching decisions and provides a unified, final ruling. In order to show the advantage of using a decision fusion deep learning method, the performance of the decision fusion network is compared to those of other common fusion methods, such as soft voting, majority voting, and averaging the decisions. For soft voting, every individual decision is assigned a probability value. The predictions are weighted by their importance and summed up. The weight that provides the highest training accuracy wins the vote.

The following section elaborates on how the neural networks are constructed for the proposed framework.

### 4.2. Vehicle Matching

The objective of the whole-vehicle matching network is to provide a similarity score between any pair of vehicle images. For this purpose, a Siamese network [23] was trained [24] using the labeled training set. Siamese network are a class of neural networks composed of two or more identical sub-networks. These sub-networks share the same parameters and weights; in addition, the weights are updated identically during training. The goal of Siamese networks is to find the similarity between the inputs through comparing their feature signatures. One advantage of a Siamese model is that the model does not have to be updated or retrained if an object class is added to or removed from the dataset. Moreover, Siamese networks are more robust to class imbalance, as a few images per class can be sufficient for training.

### 4.3. Wheel Detection

To explore the contribution of leveraging finer features from the vehicle wheels to the performance of vehicle matching, a wheel detector was trained using a portion of the training data. In particular, the wheels were manually labeled and bounded for 4077 images using LabelImg program [25]. Afterwards, the Single-Shot multibox Detection (SSD) Mobilenet v2 network [26] was retrained to detect wheels using the labeled set. The SSD Mobilenet v2 model is a single-shot detection network developed for performing object detection. The model was trained on the Common Objects in Context (COCO) image dataset [27]. The COCO dataset is geared toward large-scale object detection and segmentation. As the detection stage of the algorithm does not require special considerations, pre-trained CNNs were able to provide sufficient performance with regard to finding a bounding box around the vehicles in each video frame. The subsequently detected wheel positions were used to refine the position of the vehicle within the final cropped image; therefore, high precision is not needed in the initial bounding box coordinates. SSD MobileNet v2 was found to provide adequate precision and recall rates in detecting vehicles for this purpose; as a result, this pre-trained network was used for the detection stage for all experiments in this paper. The Mobilenet V2 model was retrained using the labeled wheel dataset. The dataset was constructed such that the wheel examples were taken from a diverse set of vehicles (e.g., sedans, SUVs, trucks, vans, and big-rigs) under different lighting conditions (e.g., day, dusk, and night). Furthermore, the wheel detector was evaluated on a part of the unlabeled testing set. In Figure 3, some examples of the detected wheels from two different vehicles in the testing set are shown.

### 4.4. Wheel Matching

As discussed earlier, the wheels can be utilized as an additional source of information; specifically, the aim is to successfully integrate the finer details from the wheel patterns into the overall vehicle matching task and thus achieve better performance in comparison to only using the whole-vehicle images by themselves. In order to match the wheels of each pair of vehicles, another Siamese network was trained to generate wheel similarity scores. The wheel detector that was developed was utilized to estimate the bounding boxes and crop the wheels from the vehicle images. The detected pairs of wheels were saved and later used to train this Siamese network. Our framework is evaluated and experimental results are shown in Section 5.

## 5. Experimental Analysis

In this section, the efficacy of our proposed fusion method is investigated, and its performance is evaluated on the dataset.

### 5.1. Vehicle Matching Results

As explained in Section 4.2, a Siamese network was trained to match any pair of images and generate a matching score. In our experiment, randomly generated image pairs were taken from a set of 543,926 images, which represent 11,918 vehicles, for training the network. True positive pairs were selected such that the two images were from two different passes of the vehicle (as opposed to two different image frames recorded during the same pass). The vehicle matching network consists of three main components. The first component has two identical branches, and each branch has seven layers. The input to each branch is a single image that has been resized to 234 × 234 × 3. The structure of each branch is depicted in Table 1. A max-pooling step is applied after each layer.

The second part of the network is the differencing phase, in which the output of one branch is subtracted from the other. The third and last component of the network is the matching network. This part of the network consists of six layers, and it operates on the differenced input to output a similarity score between 0 and 1. The structure of the matching network is listed in Table 2. These networks were implemented in Python using Tensorflow. Adaptive momentum-based gradient descent method (ADAM) technique [28] was used to minimize the loss functions and apply a learning rate of 0.005.

In our experiments, a binary cross-entropy loss function was utilized for training the network. A batch size of 256 was used for the training stage of the experiments. The network was trained for 100,000 steps, and the performance on the validation set was computed every 100 steps. In order to assess algorithm performance over the widest variety of conditions possible, a diverse subset of the validation set was constructed and used for validation. This subset contains more than 12,000 pairs of vehicle images that were collected at different locations at different times of the day and at different angles of elevation. Since the similarity score has a range of 0 to 1, a threshold of 0.5 was applied to evaluate matching accuracy. If the matching score is more than 0.5, it indicates that the two images belong to the same vehicle, and vice versa. The training and validation performance of the vehicle matching network is depicted in Figure 4. The results are also shown in Table 3.

As the threshold is varied, the accuracy value can change; thus, picking an appropriate threshold for this metric is important. In order to evaluate the performance of the vehicle matching network with different threshold values, the true positive rate (TPR), true negative rate (TNR), and matching accuracy were computed as the threshold value was varied for the test set. Accuracy here is defined as the percent of total matches performed (including any number of positive and negative pairs) for which the algorithm produced a correct answer. The resulting curves as functions of threshold value are depicted in Figure 5. From the results, it can be concluded that picking a threshold value between 0.5 and 0.8 results in near-optimal accuracy. It is worth mentioning that some applications may demand a low false negative rate, and others may demand a low false positive rate; thus, it is important to be able to select an appropriate threshold accordingly.

### 5.2. Wheel Locking

It may be of interest to note that the PRIMAVERA dataset consists of vehicle images that have been flipped, rotated, scaled, and shifted such that the front-most and rear-most wheels are always centered on the same two pixel locations in the image. Specifically, an image of size 234 × 234 × 3 was created such that the vehicle is facing to the right (determined by a tracking algorithm applied to the original image sequence) and the rear and front wheels are centered at pixels [164,47] and [164,187], respectively. This preprocessing step is here referred to as "wheel locking." The idea is that removing pose variability in the data will make it easier for the neural network to learn true discriminative features. All of the data fusion experiments in this paper used wheel-locked images as input to the whole-vehicle neural network.

An alternative to wheel locking that does not require a wheel detection step is to simply form an input image based on the bounding box returned by the initial vehicle detector algorithm. Due to the variability in how these bounding boxes are constructed, detected instances of a given vehicle may be shifted and scaled relative to one another, and any rotation of the vehicle due to sensor orientation will not be corrected.

To investigate the utility of the wheel-locking preprocessing step, the validation set performance of the whole-vehicle neural network using wheel-locked images as input was compared to when non-wheel-locked images were used as input. In both cases, image intensities were normalized to have pixel values between 0 and 1. A comparison of training and validation set accuracy between the two preprocessing methods is shown in Figure 6 and Figure 7. From the results, it can be concluded that locking the wheels location across the images enhances the matching accuracy.

### 5.3. Wheel Matching Results

In this subsection, the results of the wheel matching network are shown. The wheel detector described in Section 4.3 was employed to estimate the bounding boxes of wheels’ hubcaps from each image. The cropped wheels and their labels were used to train a Siamese network. Specifically, the network was trained using a dataset that contains 228,120 wheel images representing 11,406 unique vehicles. The dataset set was split into two sets: training (159,684 examples) and validation (68,436 examples).

All the wheel images were resized to 100 × 100 × 3. Similarly to the vehicle matching network described in Section 5.1, the wheel matching network consists of two identical branches. Each branch has four layers. The structure of each branch is tabulated in Table 4. The last layer of the Siamese network is a dense layer with 4096 neurons and a sigmoid activation function. The outputs of the two branches are then subtracted from each other and fed into the last layer, which generates the matching score. The predicted matching score is a similarity measure between 0 and 1.

The network was trained for 40,000 steps with a learning rate of 0.002. The same dataset described in Section 5.1 was used. After the wheel similarity score is computed for any pair of wheels taken from each image, a threshold is applied to the score. If the average of the two wheel matching scores is more than 0.5, then the two vehicles are declared to be the same, and vice versa. The results for the training and validation are listed in Table 5. By comparing the performances of the vehicle and wheel matching networks in Table 3 and Table 5, it can be inferred that the vehicle matching network is more reliable and accurate than the wheel matching network. This was expected because the the vehicle matching network considers the entire image and examines every aspect of the vehicle, whereas many unique vehicles may share identical or very similar wheel designs. However, it is shown later that combining the results from the vehicle and wheel networks enhances the overall matching performance.

In Figure 8, the true positive rate (TPR), true negative rate (TNR), and accuracy are shown for different thresholds.

From the figure, it can be seen that using a threshold value equal to 0.47 leads to an equal compromise between the error rates.

### 5.4. Decision Fusion Network and Results

In the following, the results of our decision fusion approach are shown. In this approach, first, all pairs of images go through the whole-vehicle matching network, which generates a whole-vehicle similarity score. Afterwards, the wheel detector described in Section 4.3 is applied to locate and crop the wheels. After at least two wheels are detected from each image, the front wheels of the two images are compared using the wheel matching network. The back wheels are also matched between the two images. As a result, three similarity scores are generated: one whole-vehicle score and two wheel scores. These steps are repeated for all the pairs of vehicles in the dataset.

As a baseline for decision fusion, the average of the three matching scores is computed. Taking the average of the scores has a reasonable chance of providing good results and is the basis of comparison for our deep fusion approach. In Figure 9, the baseline performance is evaluated by varying the threshold, similarly to the experiments done in Section 5.1. In addition, the receiver operating characteristic curves, or ROC curves, are shown in Figure 10. In Table 6, the testing matching accuracy is shown after applying a threshold value equals to 0.5 and utilizing the vehicle matching score, the wheel matching score, or both.

In addition, in Figure 11, Figure 12 and Figure 13 a comparison is provided between the baseline, vehicle, and wheel matching accuracy, true positive rate, and true negative rate for distinct threshold values. From the results, it can be concluded that combining the decisions from the vehicle and wheel matching network enhances the matching performance at some operating thresholds. The baseline approach leverages the complementary information from the wheels and provides a more accurate performance in comparison to using the vehicle images alone.

In order to substantiate our reasoning, some illustrative matching instances are shown in Figure 14, Figure 15 and Figure 16 that highlight the utility of the wheel matching. Although each pair of vehicles represents a negative match, i.e., two different vehicles, the whole-vehicle similarity scores are above the threshold, which, by themselves, indicates that they are positive matches. On the other hand, the wheel similarity scores are below the threshold, which suggests a negative match. These examples show the utility of including wheel-specific similarities along with the whole-vehicle similarity to improve matching accuracy.

To better explore the potential contribution of decision fusion in this scenario, a deep decision fusion network was constructed. Using both the vehicle and the wheel similarity scores, a multi-layered, fully connected network was trained that combines the scores and provides a final decision. This can be seen as combining the three scores by performing a smart weighted averaging of the decisions from the wheel and vehicle matching networks. Figure 17 shows the high-level block diagram for the implementation of the deep fusion network. This network is composed of four fully connected layers. The structure of the network is depicted in Table 7. The inputs to the network are the three similarity scores, and the output is the fused similarity score.

Matching scores from the original validation set were split into training (70%) and testing (30%). The new training set had 6265 pairs of vehicles, and the new testing set had 2685 pairs. The network was trained for 500 epochs. The performance of the deep fusion network was tested on the new testing set that contained 2685 pairs of vehicles. The training accuracy was 98.47%. The performance of the decision fusion network was subsequently compared to the baseline, soft voting, and majority voting performances on the testing set. The 95% confidence interval was also calculated, and the results are depicted in Table 8. This comparison shows the advantage of using a decision fusion deep learning method to smartly combine the three similarity scores as opposed to simply averaging the scores or performing soft and hard voting.

In Figure 18, a comparison among the fusion network, majority voting, and baseline matching accuracy is shown for distinct threshold values. In addition, the ROC Curves are shown in Figure 19.

## 6. Conclusions and Future Work

In this paper, a deep neural network approach was proposed for decision fusion to match pairs of vehicles through recovering feature signatures from vehicle imagery using Siamese networks. After training the vehicle and the wheel matching networks, the learned networks were then utilized to match new observed data. It was shown that leveraging pattern information specific to the wheels provided supplementary information about the vehicles in question. Experimental results showed a significant improvement in the matching accuracy after fusing the similarity scores from pairs of vehicles and their corresponding wheels’ signatures. In addition, our model performed well under diverse illumination conditions. Proposed future work includes investigating the use of vehicle imagery collected by drones for vehicle matching. Drone images may be able to provide supplementary information about the vehicles in question; for example, two similar vehicles could potentially be distinguished if their angled or elevated views reveal distinguishing differences. Moreover, on-demand data acquisition by drones would be more flexible than other traditional methods. In addition, additional sources of information could be fused, such as those describing other distinctive regions of the vehicle.

## Figures and Tables

**Figure 1 sensors-22-02803-f001:**
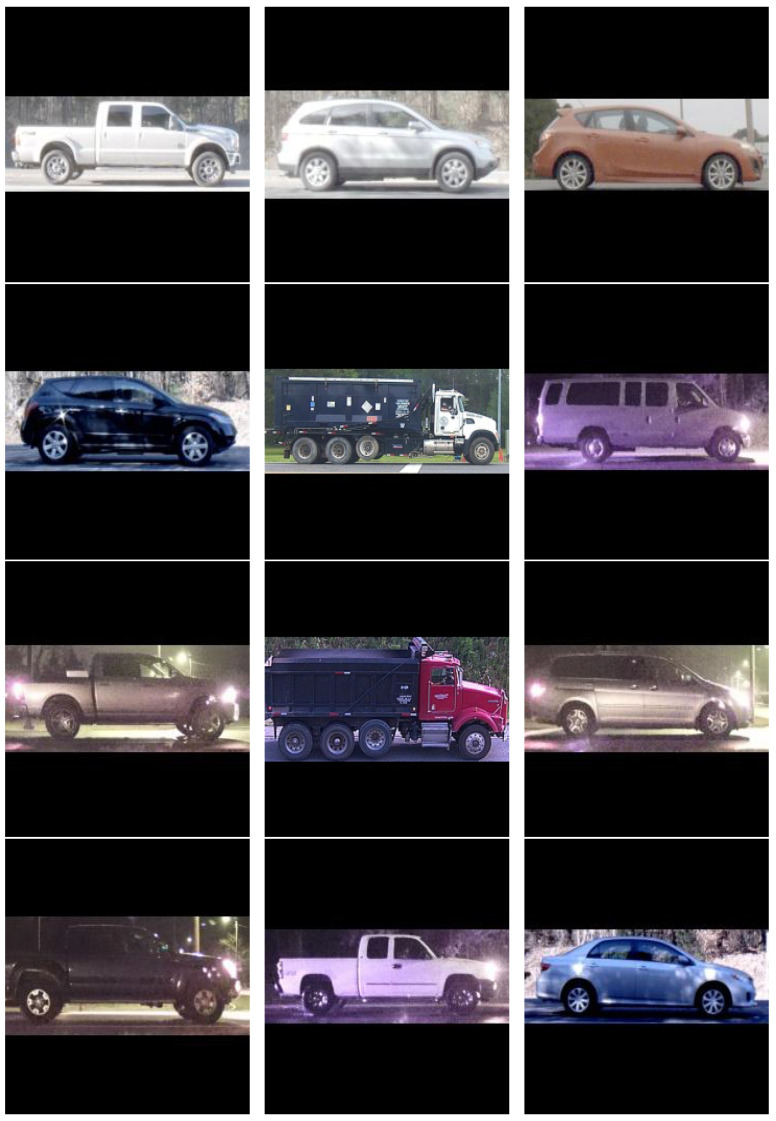
Sample images from the PRIMAVERA dataset.

**Figure 2 sensors-22-02803-f002:**
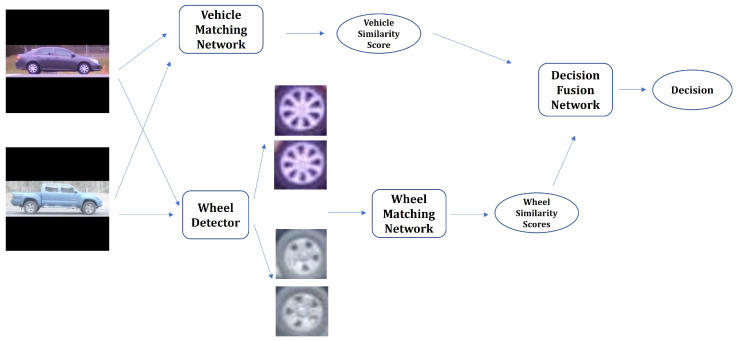
Diagram of overall vehicle matching algorithm.

**Figure 3 sensors-22-02803-f003:**
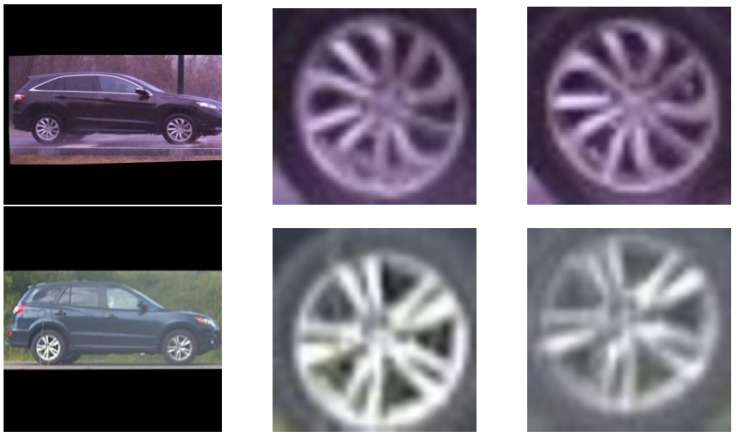
Two vehicle images and their detected wheels.

**Figure 4 sensors-22-02803-f004:**
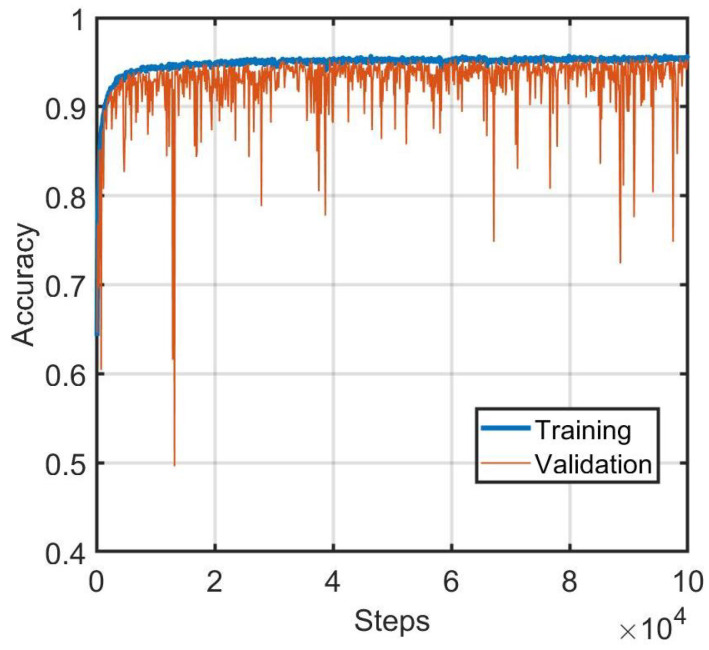
Performance of whole-vehicle matching neural network during training.

**Figure 5 sensors-22-02803-f005:**
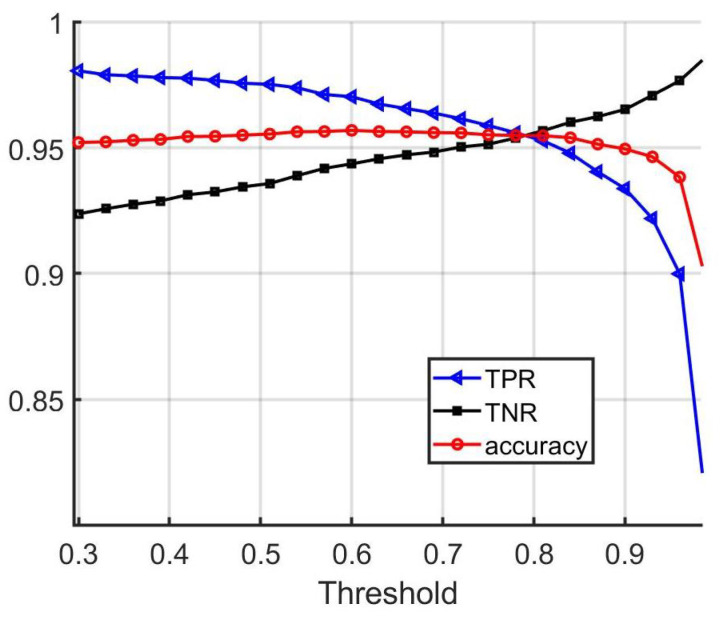
Performance of the vehicle matching network on the validation set with different threshold values.

**Figure 6 sensors-22-02803-f006:**
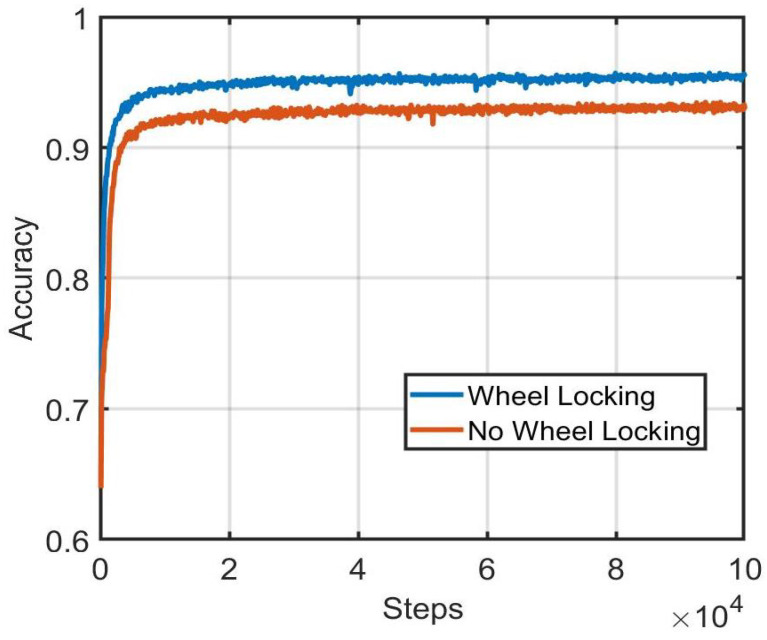
Comparison of training set performance during training between wheel-locking and non-wheel-locking preprocessing approaches.

**Figure 7 sensors-22-02803-f007:**
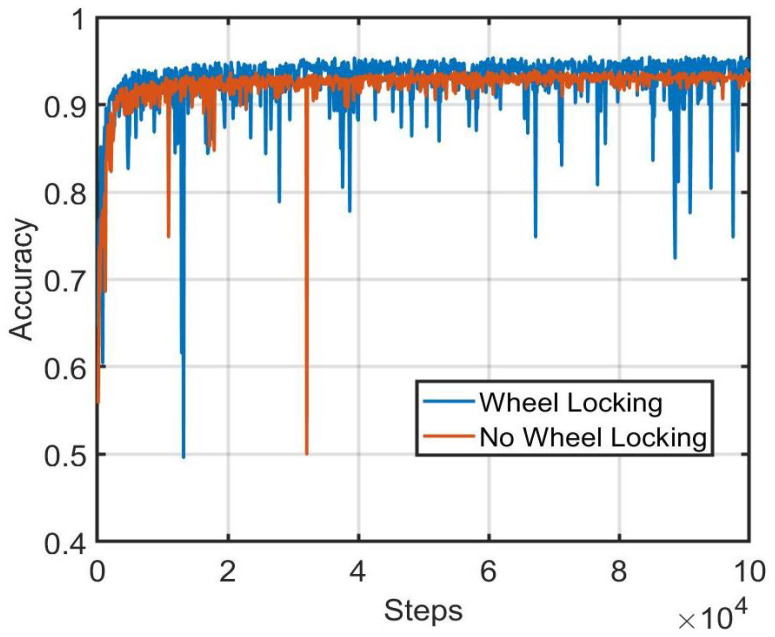
Comparison of validation set performance during training between wheel-locking and non-wheel-locking preprocessing approaches.

**Figure 8 sensors-22-02803-f008:**
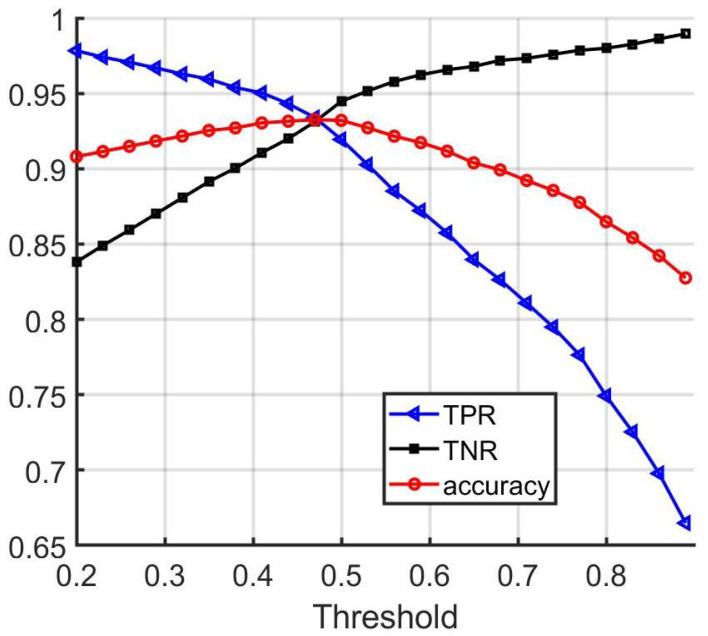
Performance of wheel matching network.

**Figure 9 sensors-22-02803-f009:**
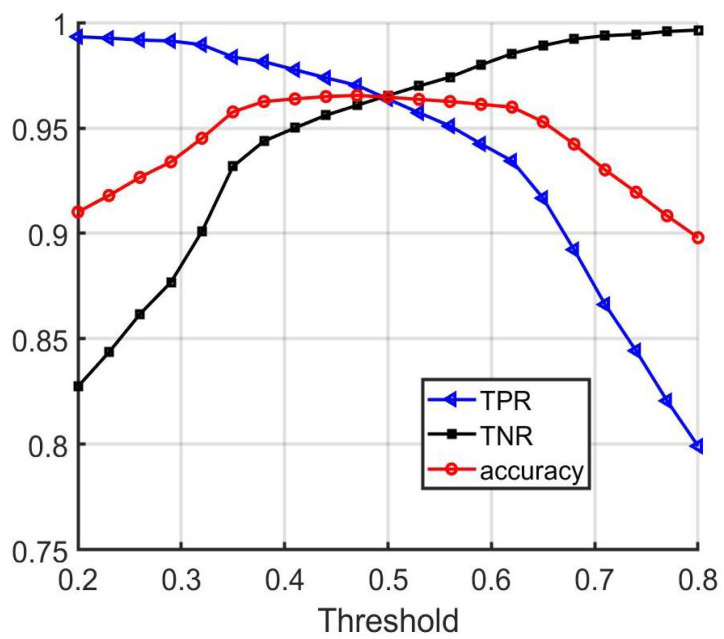
Performance of decision fusion by averaging.

**Figure 10 sensors-22-02803-f010:**
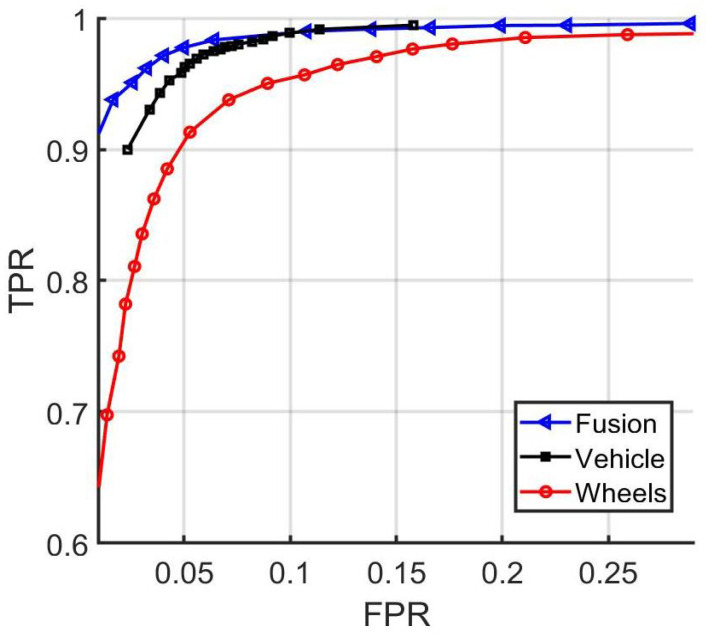
ROC curves comparing the performances of whole-vehicle-only matching, wheels-only matching, and averaging-based decision fusion of the two matching approaches.

**Figure 11 sensors-22-02803-f011:**
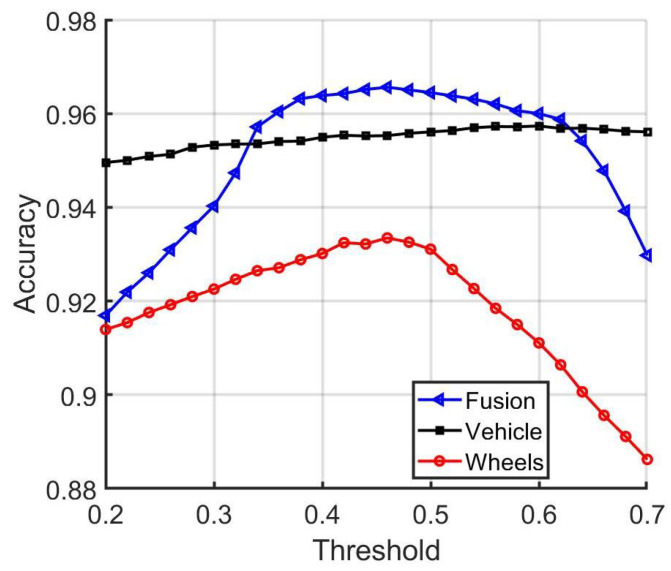
Comparison of the baseline, vehicle, and wheel-network-matching accuracies.

**Figure 12 sensors-22-02803-f012:**
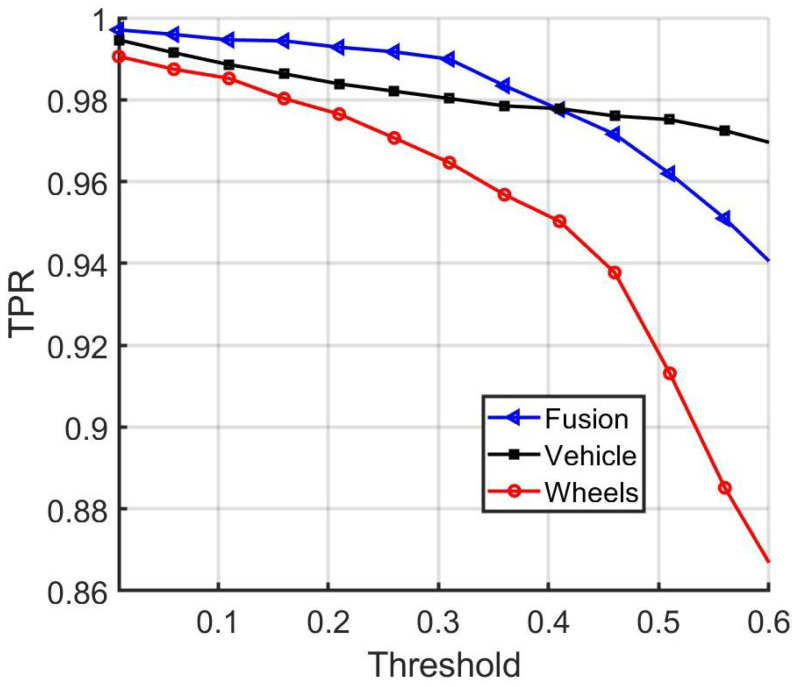
Comparison of the baseline, vehicle, and wheel-network-matching true positive rates.

**Figure 13 sensors-22-02803-f013:**
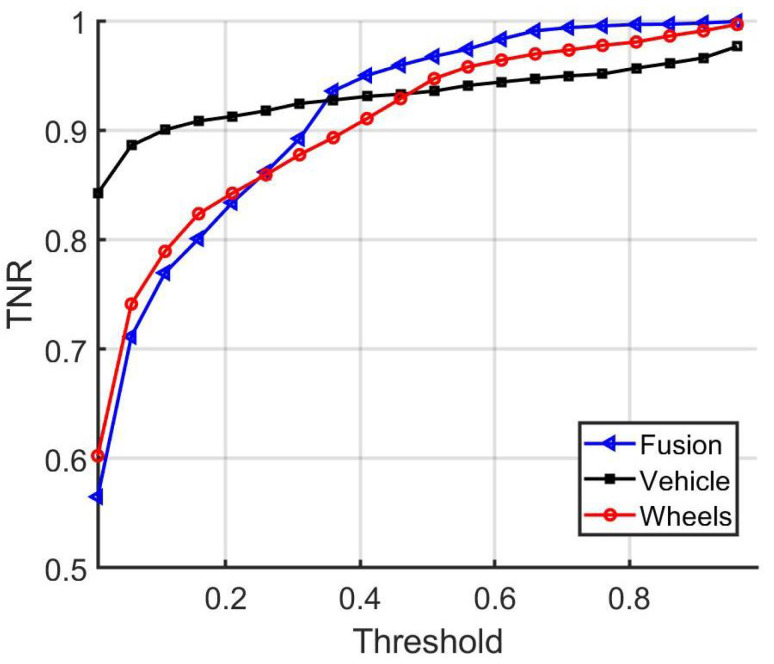
Comparison of the baseline, vehicle, and wheel-network-matching true negative rates.

**Figure 14 sensors-22-02803-f014:**
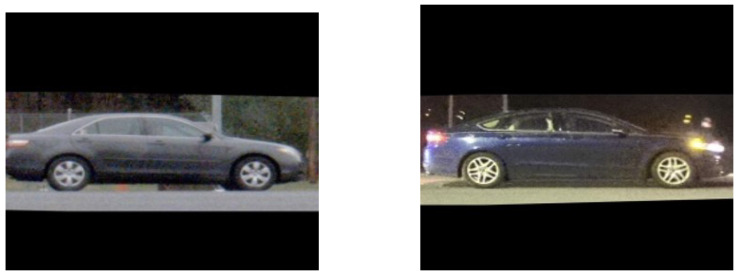
Vehicle similarity score = 0.958; wheel similarity scores = 0.01, 0.001.

**Figure 15 sensors-22-02803-f015:**
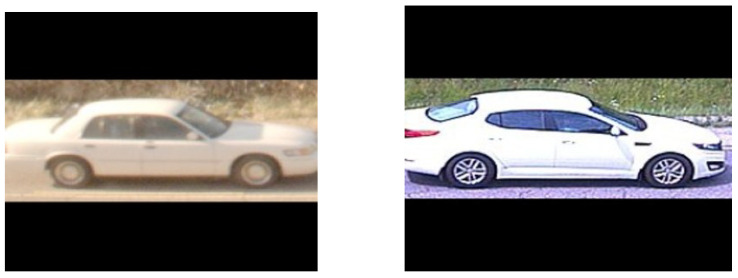
Vehicle similarity score = 0.64; wheel similarity scores = 0.02, 0.01.

**Figure 16 sensors-22-02803-f016:**
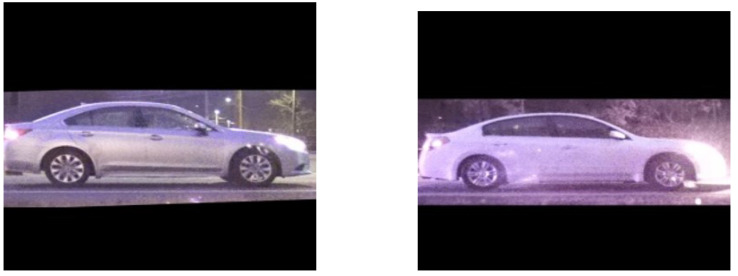
Vehicle similarity score = 0.91; wheel similarity scores = 0.01, 0.03.

**Figure 17 sensors-22-02803-f017:**
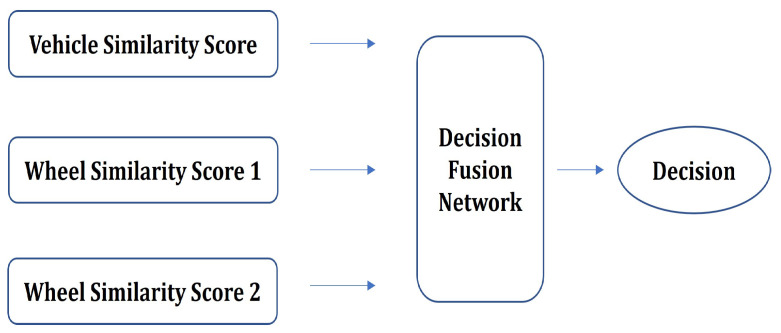
Decision fusion network.

**Figure 18 sensors-22-02803-f018:**
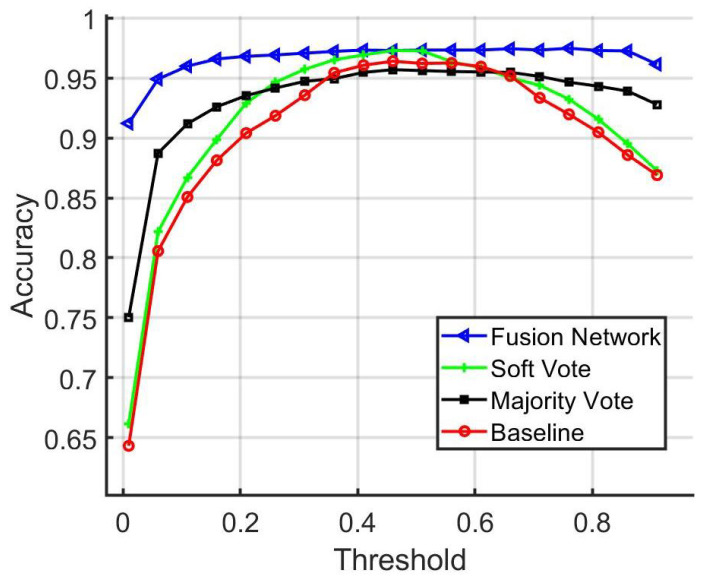
Comparison of baseline, majority vote, soft vote, and fusion network matching accuracy.

**Figure 19 sensors-22-02803-f019:**
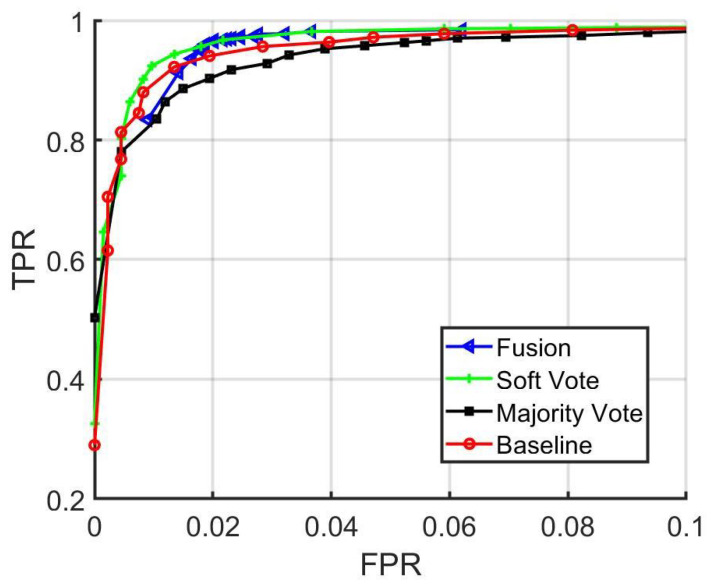
Baseline, majority vote, soft vote, and fusion network ROC curves.

**Table 1 sensors-22-02803-t001:** Vehicle matching network—each branch’s structure.

Layer	Structure	Activation
Layer 1	4 × 3 × 3	Relu
Layer 2	8 × 3 × 3	Relu
Layer 3	16 × 3 × 3	Relu
Layer 4	32 × 3 × 3	Relu
Layer 5	32 × 3 × 3	Relu
Layer 6	32 × 3 × 3	Relu
Layer 7	32 × 3 × 3	Relu

**Table 2 sensors-22-02803-t002:** Vehicle matching network—matching network structure.

Layer	Structure	Activation
Layer 1	64 × 3 × 3	Relu
Layer 2	64 × 3 × 3	Relu
Layer 3	64 × 2 × 2	Relu
Layer 4	64 × 1 × 1	Relu
Layer 5	32 × 1 × 1	Relu
Layer 6	1 × 1 × 1	Relu

**Table 3 sensors-22-02803-t003:** Vehicle matching network performance.

	Matching Accuracy
Training	95.45%
Validation	95.20%

**Table 4 sensors-22-02803-t004:** Wheel matching network structure.

Layer	Structure	Activation
Layer 1	64 × 10 × 10	Relu
Layer 2	128 × 7 × 7	Relu
Layer 3	128 × 4 × 4	Relu
Layer 4	256 × 4 × 4	Relu
Layer 5	4096 × 1	Sigmoid

**Table 5 sensors-22-02803-t005:** Wheel matching network performance.

	Matching Accuracy
Training	96.95%
Validation	93.21%

**Table 6 sensors-22-02803-t006:** Average fusion network performance on test data.

	Matching Accuracy
Vehicle Matching Score	95.5%
Wheel Matching Scores	93.21%
Average	97.63%

**Table 7 sensors-22-02803-t007:** Decision fusion network structure.

Layer	Structure	Activation
Layer 1	100	Relu
Layer 2	70	Relu
Layer 3	20	Relu
Layer 4	1	Sigmoid

**Table 8 sensors-22-02803-t008:** Comparison between fusion methods and no fusion on a portion of the testing data.

	Fusion Network	Soft Voting	Baseline	Majority Voting	Vehicle Score	Wheel Scores
Accuracy	97.77 ± 0.56%	97.28 ± 0.62%	96.31 ± 0.71%	95.68 ± 0.77%	95.46 ± 0.79%	92.93 ± 0.97%

## Data Availability

Kerekes, R.A.; Profile Images and Annotations for Vehicle Reidentification Algorithms (PRIMAVERA). Available online: http://doi.ccs.ornl.gov/ui/doi/367 (accessed on 1 January 2022), doi:10.13139/ORNLNCCS/1841347.

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
