# Peer review of "Decision-Based Fusion for Vehicle Matching"

_sensors, 2022, doi:10.3390/s22072803_

Round 1

Reviewer 1 Report

The authors proposed a framework for decision fusion utilizing features extracted from vehicle images and their detected wheels. I believe the main contribution of this paper is decision fusion. However, this part is absent in the paper. There is no clear indication of how the authors perform the decision fusion in the methodology. Except in experiments, the authors said that the average of the three matching scores is considered. More advanced techniques such as hard voting, soft voting, and data fusion can be used. I think making a decision based on two points of view is inadequate. In general, fusion is made on several points of view to be more accurate. The authors also said that they combined the three scores by performing a smart weighted averaging of the decisions from the wheel and vehicle. There is no indication about this (I mean, here where is the smart aspect). More attention should be given to the theoretical parts of the paper. In addition, more experiments should be added to compare the added value of the fusion.

Author Response

We would like to thank the reviewers for their insightful comments, which we are confident, will make the revised paper clearer. We will restate the reviewers' comments in black and provide our answers in blue. 

  • The authors proposed a framework for decision fusion utilizing features extracted from vehicle images and their detected wheels. I believe the main contribution of this paper is decision fusion. However, this part is absent in the paper. There is no clear indication of how the authors perform the decision fusion in the methodology. Except in experiments, the authors said that the average of the three matching scores is considered.

Point well taken. We have further refined the statement of the main contribution in Section 1, 2, and 4.  In this work, we investigate using neural networks to better explore the potential contribution of decision fusion. We train a multi-layered, fully connected decision fusion network that combines the matching decisions and provides a unified, final ruling. In order to show the advantage of using a decision fusion deep learning method, we compare the performance of the decision fusion network to other common fusion methods such as soft, majority voting, and averaging the decisions.

  • More advanced techniques such as hard voting, soft voting, and data fusion can be used.

Point well taken. We compared the performance of the decision fusion network to the baseline (averaging the scores), soft voting, and majority voting (hard voting) performance on the testing set in Section 5.4. The results are listed in Table 8. This comparison shows the advantage of using a decision fusion deep learning method to combine the three similarity scores as opposed to simply averaging the scores or voting.

  • I think making a decision based on two points of view is inadequate. In general, fusion is made on several points of view to be more accurate.

Point well taken. We certainly agree and for future work, we are planning to collect multi-view data so that we can address different parts of the vehicle for vehicle matching.

  • The authors also said that they combined the three scores by performing a smart weighted averaging of the decisions from the wheel and vehicle. There is no indication about this (I mean, here where is the smart aspect).

Point well taken. To better explore the potential contribution of decision fusion, we constructed a deep decision fusion network. Using both the vehicle and the wheel similarity scores, we trained a multi-layered, fully connected network that combines the scores and provides a final decision. The network learns the weights that should be assigned to each score, which could be seen as combining the three scores by performing a nonlinear weighted averaging of the decisions from the wheel and vehicle matching networks. The non-linear aspect to the weighting is a potential advantage above simpler weighting approaches. The structure of the network is listed in Section 5.4. We hope that the revisions will further clarify the network structure.

  • More attention should be given to the theoretical parts of the paper.

Point well taken. We refined the problem formulation section (Section 4.1) and we hope that the revisions will further clarify our methodology.

  • In addition, more experiments should be added to compare the added value of the fusion.

Point well taken. We compared the performance of the decision fusion network to the baseline (averaging the scores), soft voting, and majority voting (hard voting) performance on the testing set in Section 5.4. This comparison was added to show the advantage of using a decision fusion deep learning method to combine the three similarity scores as opposed to simply averaging the scores, performing voting or using either the vehicle or wheel images.

Reviewer 2 Report

The paper is about Decision Based Fusion for Vehicle Matching. I have the following comments:

  1. The Abstract is too short. It should be re-written to summarize the results
  2. The Introduction is too short. There is no motivation and there is not a single reference in the Introduction
  3. Figure 1 should be moved to a later stage when you discuss the model design
  4. In Related Work, the first reference starts with [7]. This does not male sense. Where are the first 6 references?
  5. Related work references are too old except the one that is related to the same authors of the paper. Authors are advised to search for most recent references especially from 2018 to 2022
  6. The paragraph “The balance of the paper is organized as follows ……..” should be moved to the end of Section I (Introduction)
  7. I see the reference [1] at the end of page 2. This should be the first reference
  8. The dataset should be explained in detail. How can we have about 14K unique vehicles? Give examples about unique vehicles. Is the dataset balanced or imbalanced? If imbalanced, how did you overcome this problem?
  9. The contribution of the paper is unclear. The authors used a Siamese network but what is new?
  10. There is no comparison between this work and other related work.
  11. The conclusion is very short
  12. Add a section to explain the limitations of the work and future work
  13. The references are inconsistent. They do not comply to any style. E.g. check ref [21] and [22]

Author Response

We would like to thank the reviewers for their insightful comments, which we are confident, will make the revised paper clearer. We will restate the reviewers' comments in black and provide our answers in blue.

1-The Abstract is too short. It should be re-written to summarize the results.

Point well taken. We have further refined the statement of the abstract and we hope that the revisions will further clarify the results of our work.

2-The Introduction is too short. There is no motivation and there is not a single reference in the Introduction

Point well taken. We have further improved the introduction section, and we added additional references.

3-Figure 1 should be moved to a later stage when you discuss the model design.

Point well taken.  The figure has been moved.

4-In Related Work, the first reference starts with [7]. This does not make sense. Where are the first 6 references?.

Point well taken. We adjusted the order of the references.

5-Related work references are too old except the one that is related to the same authors of the paper. Authors are advised to search for most recent references especially from 2018 to 2022

Point well taken. We added some relatively new references [1-6] in Section 1.

6-The paragraph “The balance of the paper is organized as follows ……..” should be moved to the end of Section I (Introduction)

Point well taken, we moved the paragraph to the end of Section 1.

7-I see the reference [1] at the end of page 2. This should be the first reference

Point well taken. We adjusted the order of the references.

8-The dataset should be explained in detail. How can we have about 14K unique vehicles? Give examples about unique vehicles. Is the dataset balanced or imbalanced? If imbalanced, how did you overcome this problem.

The dataset has 14K vehicles that have been identified using their license plate information, which means that we have almost 14K license plate numbers. Each vehicle (License plate) might have different number of images. However, in our experiment, we fixed the number of true matches versus false matches that we use to train the Siamese network in order to achieve a balanced dataset at training time. The dataset does contain more passenger vehicles than service-type vehicles, which may result in varying performance across different types of vehicles. For testing, we constructed a diverse subset of the validation set to use for validation throughout the paper. This subset contains more than 12,000 pairs of vehicle images that were collected at different locations at different times of the day and two different angles of elevation. The testing set is also balanced such that the number of true matches is equivalent to the number of false matches.

9-The contribution of the paper is unclear. The authors used a Siamese network but what is new.

Point well taken. We have clarified the contribution of the paper in Section 1 and we hope that the revisions will further clarify the contribution of our work. Basically, most of the work that has been done before relies on comparing three vehicle parts for detection; lights, windows, and vehicle brand, which often requires either front and back images or both for the purpose of vehicle re-identification. Moreover, they have used license plate information along with other features to re-identify the same vehicle.

In our work, we extract key features from vehicle side-view/profile images and their corresponding wheels using Siamese Networks. We believe that pattern information specific to the wheels can often provide supplementary information about the vehicles in question. We then combine the individual similarity scores derived from the extracted features to enhance overall matching accuracy; thus an overall similarity score is reached by a joint aggregation of the whole-vehicle and wheel similarity scores. To that end, we use a recently collected dataset called Profile Images and Annotations for Vehicle Re-identification Algorithms (PRIMAVERA), which we have made publicly available. Experimental results confirm a significant improvement in the vehicle matching accuracy under decision fusion.

One interesting contribution of our work is to show that “zooming in” on a known discriminative part of an object and fusing features from this part (using decision fusion) with features from the overall object can improve recognition of the object.  That is to say, a neural network trained on the object as a whole may not be as effective at learning all the discriminative parts of the object, but with user knowledge, performance can be improved by training a separate network on a specific part of the object and fusing the results.

10-There is no comparison between this work and other related work.

Point well taken. As we explained above, most of the previous work requires specific vehicle parts to be clear in the image, such as lights, windows, vehicle brand, and license plate. In this paper, we used a roadside sensor system to collect data from passing vehicles. To the best of our knowledge, this is the first work that utilizes side-view/profile images and wheels pattern for vehicle matching. We clarified that in Section 1 and Section 2, and we hope that the revisions will further clarify this point.

11-The conclusion is very short

Point well taken. We further refined the conclusion section, and we hope that the revisions will further clarify it.

12-Add a section to explain the limitations of the work and future work

Point well taken. We added a new section for future work.

13-The references are inconsistent. They do not comply to any style. E.g. check ref [21] and [22]

Point well taken. We unified the style of the references.

Reviewer 3 Report

The authors have conducted an interesting and modern research on decision fusion. A lot of work that is devoted to multi-modal data fusion, deep neural networks and decision making has been performed. The structure and research design are appropriate and fully conform to the journal topics. The authors could extend their results that are given in conclusion; however, I recommend this paper for publishing.

Author Response

- The authors have conducted an interesting and modern research on decision fusion. A lot of work that is devoted to multi-modal data fusion, deep neural networks and decision making has been performed. The structure and research design are appropriate and fully conform to the journal topics. The authors could extend their results that are given in conclusion; however, I recommend this paper for publishing.

We would like to thank the reviewers for their insightful comments, and we hope that the added revisions to section 5 (Experimental Analysis) and 7 (Conclusion) make the revised paper clearer.

Round 2

Reviewer 1 Report

The authors made an excellent effort to consider the comments suggested by reviewers. Important comments should be addressed before the final acceptance:

  • The active voice is very strong in this paper; it is preferred that the authors use passive voice.
  • The authors should provide a small justification about the use of MobileNetV2 and not other pre-trained CNN models.
  • Details about the camera (resolution, distance, …) should be added for reproducibility.
  • The authors have to define SSD before using it.
  • Conclusions and future works should be merged in one section
  • Important reference should be added to the paper:

https://doi.org/10.1002/ima.22653

Author Response

Again, we would like to thank our reviewers for their insightful comments, which we are confident, will make the revised paper clearer.

The authors made an excellent effort to consider the comments suggested by reviewers. Important comments should be addressed before the final acceptance:

1- The active voice is very strong in this paper; it is preferred that the authors use passive voice.

Point well taken. We certainly agree and we have refined our statements and used passive voice as much as we could. We hope that the revisions will further improve our statements.

2- The authors should provide a small justification about the use of MobileNetV2 and not other pre-trained CNN models.

Point well taken. Because the detection stage of the algorithm does not require special considerations, pre-trained CNNs are able to provide sufficient performance with regard to finding a bounding box around the vehicles in each video frame.  The subsequently detected wheel positions are used to refine the position of the vehicle within the final cropped image; therefore, high precision is not needed in the initial bounding box coordinates.  SSD MobileNet v2 (trained on the COCO 2017 dataset) was found to provide adequate precision and recall rates in detecting vehicles for this purpose; as a result, this pre-trained network was used for the detection stage for all experiments in this paper. We further explained this point in Section 4.3 and we hope that the revisions will further improve our justification.

3- Details about the camera (resolution, distance, …) should be added for reproducibility.

Point well taken. Three types of cameras were used for vehicle image collections.  For daytime image capture, RGB cameras equipped with either Sony IMX290 (1945x1097 pixels) or IMX036 (2080x1552 pixels) sensors were used.  For low-light image capture after sunset, a Sony UMC-S3C camera was used to perform high-sensitivity RGB imaging.  Images were captured from distances to the vehicle ranging between 1 meter and 20 meters using 1.8-mm to 6-mm lenses in conjunction with the above cameras. We included the previous statement in Section 3.1.

4- The authors have to define SSD before using it.

Point well taken. We defined SSD in Section 4.3.

5- Conclusions and future works should be merged in one section.

Point well taken. We merged both conclusions and future work in Section 6.

6- Important reference should be added to the paper:

https://doi.org/10.1002/ima.22653

Point well taken. We added the reference to the related work section (Section 2).

Reviewer 2 Report

The authors addressed my comments

Author Response

1- The authors addressed my comments

We would like to thank the reviewers again for their insightful comments, and we hope that the added revisions make the revised paper clearer.